# Neural correlates of the improvement of cognitive performance resulting from enhanced sense of competence: A magnetoencephalography study

**Takashi Matsuo, Akira Ishii✪\*, Rika Ishida, Takayuki Minami, Takahiro Yoshikawa**

Department of Sports Medicine, Osaka City University Graduate School of Medicine, Osaka City, Osaka, Japan

\* a.ishii@med.osaka-cu.ac.jp

**Data Availability Statement:** The Ethics Committee of Osaka City University (ethics@med.

## Abstract

The alterations in neural activity related to the improvement of cognitive performance, which would be leading to better academic performance, remain poorly understood. In the present study, we assessed neural activity related to the improvement of task performance resulting from academic rewards. Twenty healthy male volunteers participated in this study. All participants performed four sessions of a 1-back-Stroop task under both target and control conditions. An image indicating that the task performance of each participant was above average and categorized as being at almost the highest level was presented immediately after each session under the target condition, whereas a control image did not indicate task performance. Neural activity during the 1-back-Stroop task was recorded by magnetoencephalography. The correction rate of the 1-back-Stroop task in the final session relative to that in the first under the target condition was increased compared with the control condition. Correlation analysis revealed that the decreases in alpha band power in right Brodmann's area (BA) 47 and left BA 7 were positively associated with the increased correction rate caused by the target condition. These findings are expected to contribute to a better understanding of the neural mechanisms underlying the improvement of cognitive performance.

## Introduction

A high level of motivation is necessary to enhance physical and/or mental performance in various fields. In the field of education, motivation is one of the most important psychological concepts [1, 2], and is considered an important factor that influences student learning [3]. For example, motivation has been shown to be related to favorable educational outcomes such as enhanced curiosity, more persistence in learning, a strengthened will to learn, and the setting of higher goals [3–5]. Student motivation has also been reported to have a positive impact on academic performance [6–9].

It has been reported that academic performance is closely related to the executive functions, which is the basis of various cognitive performance [10, 11], and that the level of motivation

osaka-cu.ac.jp) which approved the protocol of our present study does not allow public sharing of the raw MEG and MRI data since the data contain potentially sensitive information. The minimal data set including the values behind the means and standard deviations, which are used to build graphs and/or figures can be received by e-mail upon request. The code for experimental task for Opensesame software was uploaded as a Supporting information file.

**Funding:** This work was supported by JSPS KAKENHI Grant No. 16H03248 (AI). The funders had no role in study design, data collection and analysis, decision to publish, or preparation of the manuscript.

**Competing interests:** The authors have declared that no competing interests exist.

for learning positively correlates with cognitive performance [11–14]. Therefore, it is of great value to clarify the neural mechanisms by which enhanced motivation to learn improves cognitive performance in order to develop effective educational methods.

It is important to note that increased motivation for learning caused by the induction of a sense of competence and achievement (i.e. academic reward) has been shown to be more important for improving educational outcomes than other forms of motivation related to external reinforcement, such as monetary rewards [4, 6, 8, 15, 16]. Furthermore, it has been reported that monetary rewards can even suppress the level of intrinsic motivation to engage in cognitive activity [17].

While there have been several studies which have investigated the neural mechanisms underlying intrinsic motivation (for example, see [18]), the neural activity associated with the improvement of cognitive performance resulting from academic rewards, even such as whether the enhancement of the neural activity in task related brain regions would occur or not, are not well understood. Therefore, we utilized the positive effects of academic rewards on the cognitive performance for the investigation of the neural mechanisms of the reward enhancing the cognitive performance in this present study. We developed a cognitive task (a 1-back Stroop task) with which the improvement of the task performance caused by the academic rewards can be evaluated.

The aim of this study was to examine whether the 1-back Stroop task we have developed is able to evaluate the alteration of cognitive performance caused by academic rewards and to clarify the neural correlates of the enhanced cognitive task performance resulting from academic rewards that induce a sense of competence. We intended to induce a sense of competence by informing the participants that their task performance was excellent during the task trials and confirmed both that their task performance was not reduced when the sense of competence was induced and that their task performance was reduced when the sense of competence was not induced (i.e., the reduction of the task performance when the sense of competence was induced was smaller than that when the sense of competence was not induced). We used magnetoencephalography (MEG) with high temporal and spatial resolutions to assess the cortical activity associated with the enhanced task performance caused by the academic reward. The neural activity corresponding to the period just before the participants' responses (i.e., when pressing the designated buttons) in each trial of the 1-back Stroop task was assessed using a spatial filtering method to detect changes in oscillatory brain activity reflecting changes in neural dynamics [19–21].

## Materials and methods

### Ethics statements

The Ethics Committee of Osaka City University approved the study protocol (approval No. 4107). Each participant provided written informed consent to participate in this study and all methods were performed in accordance with the Declaration of Helsinki and the Ethical Guidelines for Medical and Health Research Involving Human Subjects in Japan (Ministry of Education, Culture, Sports, Science and Technology and Ministry of Health, Labour and Welfare).

### Participants

Twenty healthy male volunteers aged 21.5 ± 1.2 years (mean ± standard deviation [SD]) participated in this study. All participants were confirmed to be right-handed according to the Edinburgh Handedness Inventory [22]. In addition, before starting the experiments, each participant was asked in the form of a questionnaire about the presence of present illness, symptoms, and/or health problems. None of the participants reported having any health problems

such as neural disorders, brain injury, or a history of mental illness. Current smokers and individuals taking chronic medications that affect the central nervous system were excluded.

## Experimental design

The experiment consisted of target and control conditions. Each condition was performed on the same day in a two-crossover design (Fig 1A). In the target and control conditions, the participants lay in a supine position on a bed placed in a magnetically shielded room and were asked to perform a 1-back Stroop task during MEG recording. In the 1-back Stroop task, one of four words (Red, Blue, Yellow, or Green in Japanese Kanji pictograms) was displayed in one of four font colors (red, blue, yellow, or green) for a maximum of 500 ms. The visual angle of the Kanji pictograms projected on the screen was 5.7˚ × 5.7˚ (horizontal × vertical). The word and the color of the word were selected at random for each presentation (Fig 1B). The participants were asked to judge whether the font color of the word presented at the time was the same as that of the previous one and to press the right (i.e., the same) or left button (i.e., not the same) with their right index or middle finger, respectively. As soon as they pressed the button, whether their response was correct was briefly displayed so that the time between the start of the presentation of the present word and that of the next was 650 ms. An MEG-compatible response device (HHSC-164-L; Current Designs, Philadelphia, PA) was used to record the responses, and a projector (PG-B10S; SHARP, Osaka, Japan) was used to display the words on the screen. Visual stimuli were presented by using OpenSesame software (Version 3.1.7-py2.7-win32; [23]). One set of the 1-back Stroop task consisted of 560 trials (i.e., 560 words), and four sets of the task were performed under the target and control conditions. The participants were instructed to perform the task as quickly and correctly as possible.

Under the target condition, an image indicating that the performance of each individual in the 1-back Stroop task was above average and categorized as being at almost the highest level was presented for 10 s between tasks 1 and 2, tasks 2 and 3, tasks 3 and 4, and after the task 4 (Fig 1C). This image was presented to induce a sense of competence as an academic reward. The participants were told that their task performance was assessed based on not only the percentage of correct answers, but also the response time; this information had not been previously provided. However, in fact, the task performance shown to the participants was not based on their actual task performance; the image always indicated that their task performance was categorized as being at almost the highest level.

Under the control condition, an image indicating the temperature of the fingers of the right hand of each individual was presented for 10 s between tasks 1 and 2, tasks 2 and 3, tasks 3 and 4, and after the task 4 (Fig 1D). The participants were told that the button system used in the 1-back Stroop task could also measure the temperature of the fingers. However, in fact, the temperature of the fingers was not assessed during the experiments, and the image indicating the temperature of the fingers was the same for every participant and the same during the course of the task. In order to make the participants believe that the temperature of their fingers was actually measured, they were instructed that the bar indicating their temperature corresponded to some range of temperature although the scale and the unit were not shown and therefore, the image indicating the temperature of the fingers might be the same during the course of the experiment even if slight fluctuation of the temperature existed.

At the beginning of the experiment, the participants practiced the 1-back Stroop task until they could perform the task properly. A 5-min rest period, during which participants'

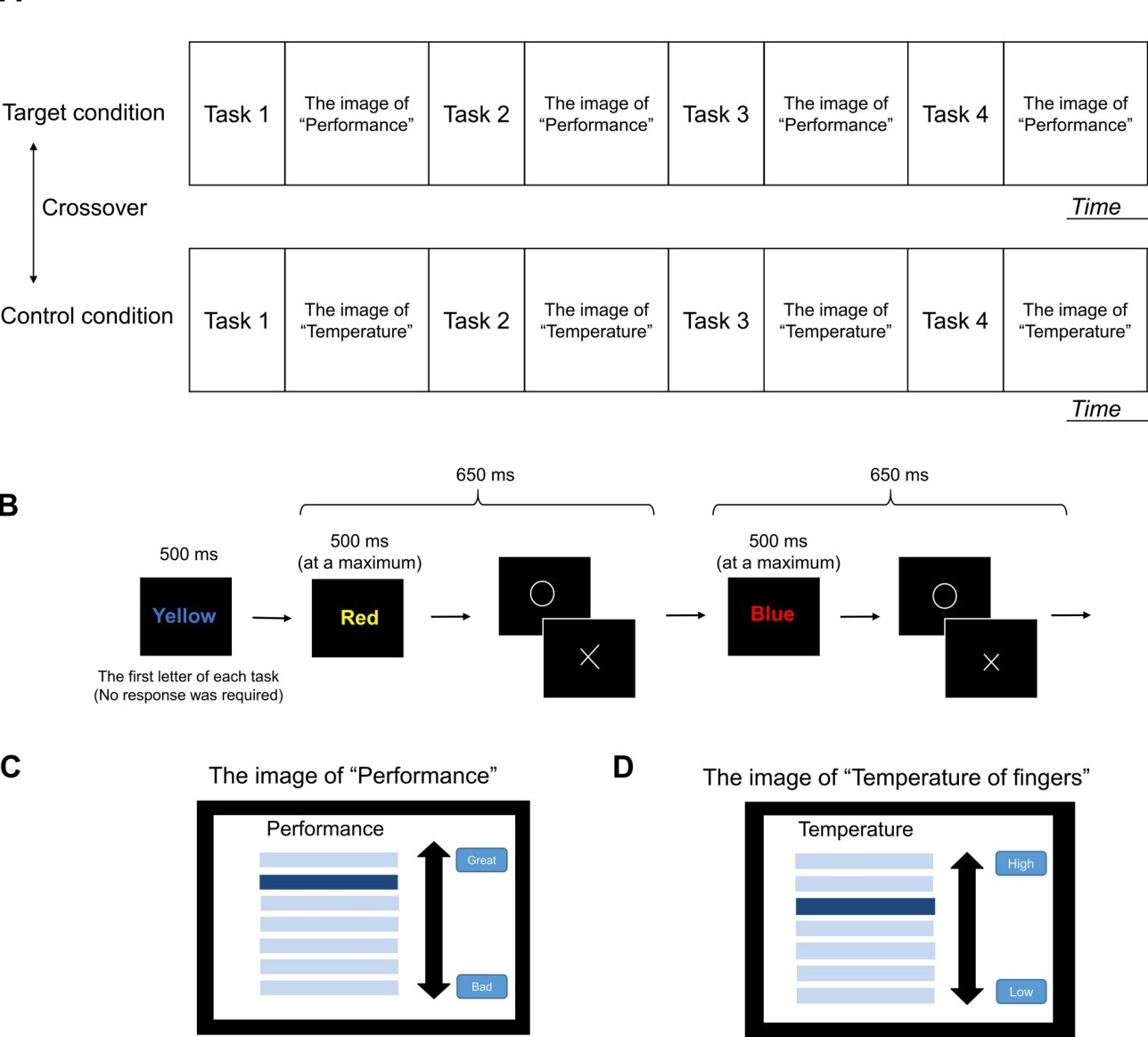

**Fig 1. Experimental design.** (A) The experiment consisted of target and control conditions, and each condition was performed on the same day in a two-crossover design. The participants were asked to perform a 1-back Stroop task four times under both the target and control conditions. (B) In the 1-back Stroop task, one of four words (i.e., Red, Blue, Yellow, and Green in Japanese Kanji pictograms) written in one of four font colors (i.e., red, blue, yellow, and green) was presented for a maximum of 500 ms. The participants were asked to judge whether the font color of the word presented at the time was the same as that of the previous one and to press either the right (i.e., the same) or left button (i.e., not the same) with their right or middle finger, respectively. As soon as they pressed the button, whether their answer was right or wrong was displayed for some milliseconds so that the time between the start of the presentation of the present word and that of the next word was 650 ms. One set of the 1-back Stroop task consisted of 560 trials (i.e., 560 words). (C) An image indicating that the task performance of each participant was above average and categorized as being at almost the highest level (i.e., "Performance") was presented for 10 s immediately after each session under the target condition. (D) A control image not indicating task performance (i.e., "Temperature of fingers") was presented for 10 s under the control condition.

heads were outside of the MEG helmet, was instituted between the target and control conditions.

Subjective levels of motivation were assessed using a visual analog scale (VAS) displayed on the screen just before task 1 and after each presentation of "Performance" or "Temperature" in the target and control conditions, respectively.

## MEG recording

As originally described in Ishii et al. [24], MEG was recorded using a 160-channel whole-head-type MEG system (MEG Vision; Yokogawa Electric Corporation, Tokyo, Japan) with a magnetic field resolution of 4 fT/Hz$^{1/2}$ in the white noise region. The sensor and reference coils were gradiometers with a 15.5 mm diameter and 50 mm baseline, and the two coils were separated by 23 mm. The sampling rate was 1,000 Hz and the obtained data were high-pass filtered at 0.3 Hz and low-pass filtered at 500 Hz.

## MEG analyses

The analyses of the MEG data were performed using a similar method to that described in a previous study [24]. Magnetic noise from outside the magnetically shielded room was eliminated by subtracting the data obtained from the reference sensors using a software program (MEG 160; Yokogawa Electric Corporation). The reference sensors were located in the liquid helium dewar, away from participants' head, in the magnetically shielded room. Epochs of the MEG data, including artifacts, such that originated from outside the magnetically shielded room remaining after the elimination using reference sensors and physiological noises caused by eye blinks and so on, were identified visually and excluded from the analysis. The total length of the epochs was 250 ms from -300 to -50 ms before the button press as described below. In addition, electro-oculography (EOG) was recorded, from electrodes placed over the left and right ends of the eyebrow of the left eye by using Neurofax 1000 (Nihon-Kohden, Tokyo, Japan) with sampling rate of 1000 Hz applying 0.5 Hz low-cut filter and 120 Hz high-cut filter, over the MEG recordings to help identify the physiological noises caused by eye blinks and eye movements. The data obtained by empty room recordings were not used. Spatial filtering analysis of the MEG data was performed to identify changes in oscillatory brain activity reflecting the time-locked cortical activities caused by performing the 1-back Stroop task. The MEG data were bandpass filtered at 8–13 Hz using the finite impulse response filtering method implemented in Brain Rhythmic Analysis for MEG software (BRAM; Yokogawa Electric Corporation).

After the bandpass filtering, the location and intensity of the cortical activities were estimated using BRAM, which uses a narrow-band adaptive spatial filtering algorithm [25, 26]. The adaptive spatial filtering method that implemented in the software was a lead-field normalized version of a linearly constrained minimum variance filtering method [27] and the procedure was optimized for time-frequency source reconstruction [25]. The mathematical details of the source reconstruction performed in our present study are present in the previous report [28]. The voxel size was set at $5.0 \times 5.0 \times 5.0$ mm. Under both the target and control conditions, the oscillatory power of the MEG data for task 4 in a 250-ms time window (from –300 to –50 ms before the button press) was calculated relative to that of task 1. The parameter settings of the software were as follows: The active period was "-300 to -50 ms" before the button press in task 4, the baseline period was "-300 to 500 ms" before the button press in task 1, the functional type was {"ERD", "ERS"}, the functional unit was "F-ratio [dB]", the voxel spacing was "5 mm", and the voxel range was "inside the spherical conductor model".

The MEG data analyzed using the spatial filtering method were further analyzed using statistical parametric mapping (SPM8, v6313; Wellcome Department of Cognitive Neurology, London, UK) implemented in Matlab (ver. 7.11.0; Mathworks, Natick, MA). The MEG parameters were transformed into the Montreal Neurological Institute T1-weighed image template. The anatomically normalized MEG data were filtered with a Gaussian Kernel of 20 mm (full-width at half-maximum) in the x-, y-, and z-axes. The individual MEG data were

then incorporated into a random-effects model. The parameters estimated were used to create "contrast" images for the group analyses. The contrast of the alteration in oscillatory power corresponding to [(Task 4 / Task 1 in the target condition)–(Task 4 / Task 1 in the control condition)] was calculated using the Image Calculator implemented in SPM8. To assess the neural activity correlated with the alteration of the correction rate caused by the academic reward under the target condition, the contrast images were analyzed by applying a one-sample $t$-test with the alteration in the correction rate (i.e., the correction rate in task 4 under the target condition relative to that under the control condition) as a covariate. The threshold for the analysis was set at $P < 0.025$ (family-wise error corrected for multiple comparisons), considering the multiple comparisons (i.e., the increase or decrease in alpha band power in task 4 compared with that in task 1). However, the statistical threshold for the statistical parametric map shown in our figure was set at $P < 0.05$ (family-wise error corrected for multiple comparisons) for the purpose of presentation. Localization of the brain regions was performed using WFU_PickAtras, Version 3.0.4 (http://fmri.wfubmc.edu/software/pickatlas) and Talairach Client (Version 2.4.3; http://www.talairach.org/client.html). As an additional analysis, the correlation analysis of the neural activity with the alteration of the subjective motivation was performed: The statistical threshold for this additional analysis was also set at $P < 0.025$.

## Overlay of Magnetic Resonance (MR) image

As originally described in Ishii et al. [24], anatomical magnetic resonance imaging (MRI) was performed for each participant using a Philips Achieva 3.0 TX scanner (Royal Philips Electronics, Eindhoven, the Netherlands) to permit registration of magnetic source locations with their respective anatomical locations. Five markers (Medtronic Surgical Navigation Technologies, Inc., Broomfield, CO) were attached to the scalp (i.e., two markers 10 mm in front of the left and right tragus, one marker 35 mm above the nasion, and two markers 40 mm to either side of the marker above the nasion). The MEG data were superimposed on the MR images using information obtained from these markers and the MEG localization coils, which were attached to the scalp over the MEG recording at just the same points as those at which MR-markers were attached, using the MEG 160 software. Based on the empirical evidence reported in the literature [29] suggesting that the negative effects caused by an MRI scan on MEG data typically disappear within 3 days after the MRI scan, the MEG study was planned to be performed at least 4 days after the MRI study to avoid effects caused by magnetization if the MEG study were not able to be performed before the MRI study. In fact, the interval between the MEG study and the MRI scan was more than 5 days in our present study when the MRI scan preceded the MEG study.

## Statistical analysis

All values are presented as mean ± SD unless otherwise stated. A paired $t$-test corrected for multiple comparisons was used to compare the level of motivation between before and after task 1, before task 1 and after task 2, before task 1 and after task 3, and before task 1 and after task 4, and the correction rate between tasks 1 and 2, tasks 1 and 3, and tasks 1 and 4, under both the target and control conditions (i.e., the $P$ values of the paired $t$-test for the motivation were multiplied by 8 and those for the correction rate was multiplied by 6). Effect size of each paired t-test was calculated by using G*Power software (Version 3. 1. 9. 2; [30]). All $P$ values were two-tailed, and values $< 0.05$ were considered statistically significant for the tests without multiple comparisons.

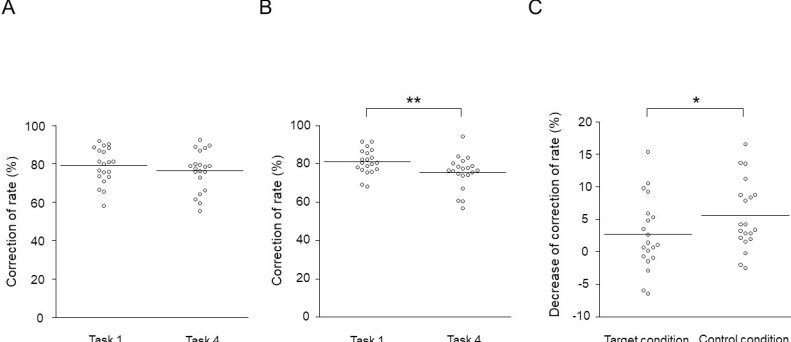

**Fig 2. Correction rates for the 1-back Stroop tasks.** (A) The correction rates for tasks and 4 under the target condition. (B) The correction rates for tasks 1 and 4 under the control condition. (C) The decreases in the correction rate for task 4 compared with that for task 1 under the target and control conditions. The horizontal lines indicate mean values. $^{**}P < 0.01$ and $^{*}P < 0.05$.

## Results

### Correction rates for the 1-back Stroop tasks

The correction rate for task 4 under the target condition was not altered compared with that for task 1 under the target condition ($t_{19} = 2.141$, d = 0.51; $P = 0.272$, paired $t$-test with the Bonferroni correction; Fig 2A) and the correction rate for task 4 under the control condition was decreased compared with that for task 1 under the control condition ($t_{19} = 4.642$, d = 1.06; $P = 0.00106$, paired $t$-test with the Bonferroni correction; Fig 2B). In addition, the decreased correction rate for task 4 compared with that for task 1 under the target condition was smaller than that under the control condition ($t_{19} = 2.176$, d = 0.50; $P = 0.0430$, paired $t$-test without the Bonferroni correction; Fig 2C).

### Motivation

Subjective levels of motivation were analyzed for 19 participants whose VAS data were properly saved. The subjective level of motivation after task 4 under the target condition was decreased compared with that just before task 1 under the target condition ($t_{18} = 3.526$, d = 0.83; $P = 0.0160$, paired $t$-test with the Bonferroni correction; Fig 3A). The subjective level

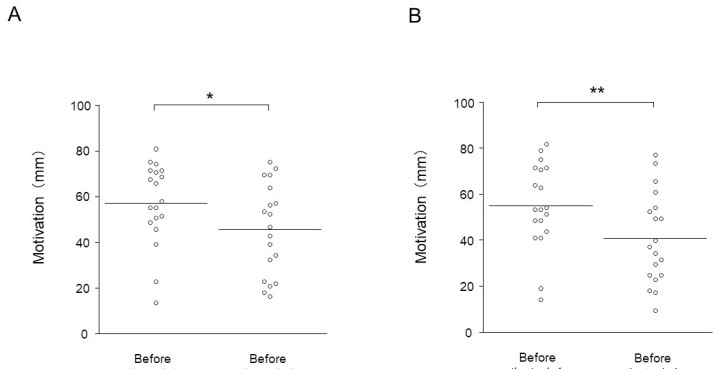

**Fig 3. Subjective levels of motivation.** (A) The subjective levels of motivation before task 1 and after task 4 under the target condition. (B) The subjective levels of motivation before task 1 and after task 4 under the control condition. The horizontal lines indicate mean values. $^{*}P < 0.05$ and $^{**}P < 0.01$.

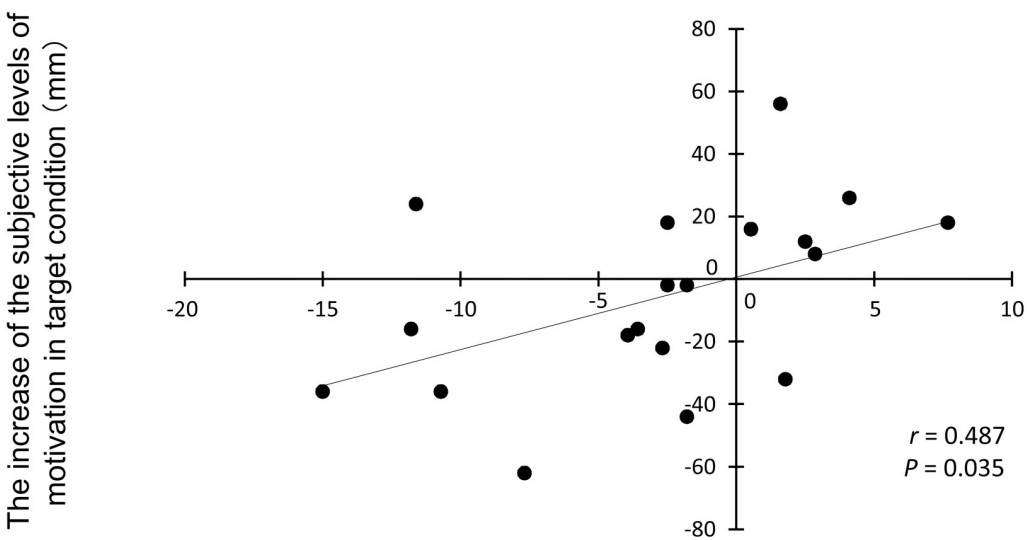

**Fig 4. Relationship between the increase in the correction rate under the target condition and the increase in subjective levels of motivation under the target condition.** The linear regression line, Pearson's correlation coefficient, and the *P* value are shown.

of motivation after task 4 under the control condition was decreased compared with that just before task 1 under the control condition ($t_{18}$ = 5.352, d = 1.26; $P$ = 0.000320, paired *t*-test with the Bonferroni correction; Fig 3B). The subjective level of motivation after task 4 under the target condition showed a tendency to increase compared with that under the control condition ($t_{18}$ = 1.961, d = 0.46; $P$ = 0.0650, paired *t*-test without the Bonferroni correction), and the increase in the subjective level of motivation in task 4 under the target compared with the control condition was positively associated with the increase in the correction rates in task 4 under the target compared with the control condition (r = 0.487, $P$ = 0.0350, Fig 4). The alteration of the neural activity that correlated with the alteration of the subjective motivation were not observed.

## Spatial filtering analysis of the MEG data

The MEG data from four participants were excluded (i.e., the MEG data were analyzed for 16 participants) because the numbers of the trials of their MEG data after the removal of the trials contaminated with artifacts were not sufficient for the analysis (i.e., the number of the trials of the MEG data < 40 epochs were excluded from our analysis). The correlation between the changes in neural activity during task 4 under the target compared with the control condition, and the increase in the correction rate of task 4 under the target compared with the control condition was assessed. As a result, the decrease of alpha band power in Brodmann's area (BA) 47 and 7 was positively correlated with the increase of the correction rate (Fig 5; Table 1).

## Discussion

In the present study, the participants performed four sessions of a 1-back Stroop task under a target and control condition. An image that was intended to induce a sense of competence and achievement was presented at the end of each session under the target condition, whereas an

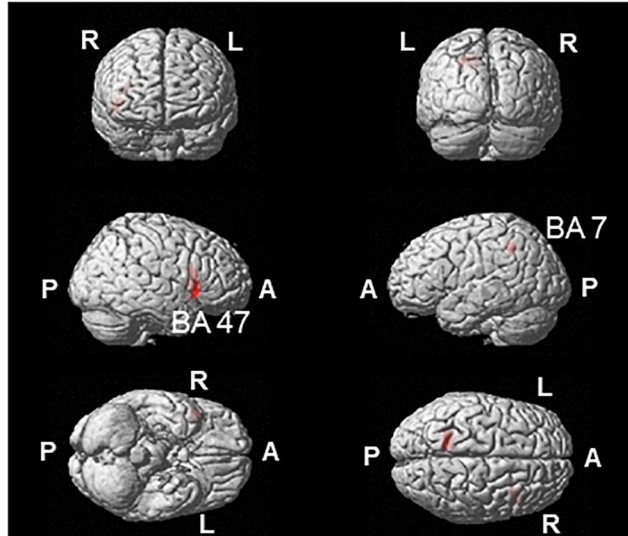

**Fig 5. Statistical parametric map of the brain regions where a decrease in alpha band power was positively correlated with an increase in the correction rate under the target condition.** Random-effects analysis of 16 participants; $P < 0.05$, family-wise error rate corrected for the entire search volume. R, right; L, left; A, anterior; P, posterior.

image that did not induce a sense of competence was presented under the control condition. The correction rate for the final session (i.e., task 4) was decreased compared with that for the first session (i.e., task 1) under the control condition, but not under the target condition. The decrease in the correction rate for task 4 relative to that for task 1 was suppressed under the target compared with the control condition. In addition, the subjective level of motivation tended to increase under the target compared with the control condition, and the increase in the subjective level of motivation observed under the target condition was positively associated with the increase in the correction rate in the final task under the target compared with the control condition.

We intended to adopt a cognitive task capable of detecting minute alterations in task performance caused by the induction of a sense of competence and develop a type of 1-back Stroop task (i.e., the 1-back Stroop task used in this study). As a result, since the correction rate of the final session was decreased compared with that of the first session only under the control condition, the 1-back Stroop task seemed to be able to detect the alterations in task performance induced by the presentation of an image intended to induce a sense of

**Table 1. Correlations between alterations in alpha band oscillatory brain activity and the increase in the correction rate under the target relative to the control condition.**

| Decrease/increase | Frequency | Location | BA | MNI coordinate(mm) | | | Z value |
|---|---|---|---|---|---|---|---|
| | | | | x | y | z | |
| Decrease | 8–13 Hz | Inferior Frontal Gyrus | 47 | 47 | 18 | 0 | 4.06 |
| Decrease | 8–13 Hz | Precuneus | 7 | -18 | -52 | 45 | 4.00 |

BA, Brodmann area; MNI, Montreal Neurological Institute.

x, y, z: Stereotaxic coordinate.

Data were obtained from random-effect analysis. Only significant change is shown (paired *t*-test, $P < 0.025$, corrected for multiple voxel-wise comparisons with family-wise error rate).

competence. In addition, since the increase in the subjective level of motivation observed under the target compared with the control condition was positively associated with an increase in the correction rate in the final task observed under the target condition, it can be speculated that the increased motivation caused by the presentation of an image intended to induce a sense of competence enhanced the cognitive performance of the 1-back Stroop task in this study. Since the 1-back Stroop task is a combination of Stroop and n-back tasks, the performance of a 1-back Stroop task can be considered to be related to cognitive function, such as selective attention [31, 32] and working memory [33], which are typically categorized within executive function [10, 11, 34, 35].

In the present study, neural activity during the 1-back Stroop task was assessed using MEG, and a correlation analysis between the alterations in neural activity and the increased task performance induced under the target condition was performed. Since alpha band oscillatory brain activity has been reported to be related to cognitive function and performance, we focused on alterations in alpha band oscillatory brain activity: A decrease in alpha band power has been reported to be related to the activation of cortical areas involved in sensory or cognitive information [36], and that high cognitive and memory performance is related to event-related decreases in alpha band power [37, 38]. We found that the decreases in alpha band power in right Brodmann's area (BA) 47 and left BA 7 just before the button press were positively correlated with the increase in the correction rate caused by the target condition. It has been reported that the inferior frontal gyrus (also referred to as the ventrolateral prefrontal cortex), including BA 47, is involved in working memory [39], and the activation of BA 47 has been observed while performing an n-back task [33, 40]. In fact, the level of activation in the right inferior frontal gyrus has been reported to be positively related to the performance of a n-back task in a functional MRI (fMRI) study [41]. In addition, the activation of the inferior frontal gyrus has been reported to be related to performance in a Stroop task [42, 43]. BA 7 has also been found to be related to performance in an n-back task [33, 44, 45] and other kinds of working memory tasks [46]. Therefore, it is suggested that the improved performance in the 1-back Stroop task caused by the academic reward was due to the activation of these task-related brain regions under the target condition.

In addition to being a task-related brain region, BA 7 may play a key role in modulating cognitive task performance through motivation (i.e., the BA 7 may be the brain region linking the enhanced motivation to the improved performance). A recent fMRI study showed that the left BA 7 was connected with the whole task-related brain regions during the performance of the symbol digit modalities test and that the BA 7 works as an in-between hub connecting the task related brain regions to other brain regions in the default mode network [47], parts of which were involved in signaling of reward outcomes [48] and achievement motivation [49]. In fact, it has been reported that oscillatory brain activity in BA 7 assessed by MEG was associated with increased motivation to perform forthcoming cognitive task [50]. Taking theses into consideration, it can be hypothesized that the BA 7 is a key brain region in enhancing cognitive performance through motivation. To test this hypothesis, further studies on the functional relationships between these brain regions are necessary.

This study did have several limitations. First, all the participants were healthy adult males. To generalize our findings to actual academic situations, studies involving females and/or children and adolescents are needed. Second, some neural activity related to the increase in motivation caused in the present study might have been missed, as MEG has relatively low signal-to-noise ratio for signals originating from deep brain regions, such as the striatum, compared to that form the cortical regions. Third, since we were interested in the neural mechanisms related to the improvement of cognitive performance and it has been reported that academic rewards can improve academic performance, we focused on the effects of an academic reward

on cognitive performance and did not assess whether the cognitive performance of the 1-back Stroop task used in our study was enhanced or deteriorated by other types of rewards such as monetary rewards. Fourth, since we intended to test whether the improvement of cognitive performance would be caused by the performance related feedback which would work as an academic reward that induce a sense of competence, we did not examine whether the improvement of cognitive performance would be caused by the performance related feedback which would not work as academic rewards such that showing images indicating that the performance of each individual in the 1-back Stroop task was equal to or below the average in our present study. It is of interest to examine whether the improvement of cognitive performance is also caused by showing images indicating that the performance of each individual is equal to or below the average in future researches. Fifth, we performed source reconstruction of the MEG data because we wanted to clarify precise brain regions correlated with the enhanced cognitive task performance resulting from academic rewards. Since the sensor level analysis of MEG data is sensitive to the head position in the whole-head MEG helmet and the position of the head in the MEG helmet was not precisely the same between the participants and even within each participant between the conditions because the participants' heads were out of the MEG helmet during the rest period between the conditions in our present study, we think that the MEG analysis with source reconstruction is adequate in our present study.

In conclusion, we found that viewing an image intended to induce a sense of competence (i.e., an academic reward) increased cognitive performance and motivation, and that BA 47 and BA 7 were related to the increase in cognitive performance caused by the enhanced motivation induced by viewing the image. Our finding that cognitive performance can be increased by academic rewards through enhanced activation of task related brain regions would motivate further studies to clarify the neural mechanisms by which academic performance is improved by enhanced motivation to learn in longer time span, leading to establish scientific basis for the educational methods aimed to enhance motivation to learn and to devise effective educational methods.

## Supporting information

**S1 File. Cade for experimental task.** Code for the 1-back Stroop task in Opensesame format. (ZIP)

## Acknowledgments

We wish to thank Forte Science Communications for editorial help with the manuscript and Manryoukai Imaging Clinic for assistance with the MRI scans. We also wish to thank Mr. Touki Kobuchi and Mr. Kiyotaka Sumi for help in performing our experiment.

## Author Contributions

**Conceptualization:** Takashi Matsuo, Akira Ishii.

**Data curation:** Takashi Matsuo, Akira Ishii.

**Formal analysis:** Takashi Matsuo, Akira Ishii.

**Funding acquisition:** Akira Ishii.

**Investigation:** Takashi Matsuo, Akira Ishii, Rika Ishida, Takayuki Minami.

**Methodology:** Takashi Matsuo, Akira Ishii.

**Project administration:** Akira Ishii.

**Software:** Akira Ishii.

**Supervision:** Akira Ishii, Takahiro Yoshikawa.

**Validation:** Akira Ishii.

**Visualization:** Akira Ishii.

**Writing – original draft:** Takashi Matsuo, Akira Ishii.

**Writing – review & editing:** Akira Ishii.

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
