## [Decision Letter · Decision Letter 0]

18 Mar 2021

PONE-D-21-01757

Neural correlates of the improvement of cognitive performance resulting from enhanced sense of competence: a magnetoencephalography study

PLOS ONE

Dear Dr. Ishii,

Thank you for submitting your manuscript to PLOS ONE. After careful consideration, we feel that it has merit but does not fully meet PLOS ONE’s publication criteria as it currently stands. Therefore, we invite you to submit a revised version of the manuscript that addresses the points raised during the review process.

The reviewers bring up a number of suggested revisions that can likely be dealt with by the authors. I will also warn that if the justification for male only participants is not strong, I will require for female participants to be added in order to agree to publishing this work in PLOS One.

We look forward to receiving your revised manuscript.

Kind regards,

Desmond J. Oathes

Academic Editor

PLOS ONE

Journal Requirements:

Reviewers' comments:

Reviewer's Responses to Questions

**Comments to the Author**

1. Is the manuscript technically sound, and do the data support the conclusions?

Reviewer #1: Yes

Reviewer #2: No

2. Has the statistical analysis been performed appropriately and rigorously? 

Reviewer #1: Yes

Reviewer #2: No

3. Have the authors made all data underlying the findings in their manuscript fully available?

Reviewer #1: Yes

Reviewer #2: No

4. Is the manuscript presented in an intelligible fashion and written in standard English?

Reviewer #1: Yes

Reviewer #2: Yes

5. Review Comments to the Author

Reviewer #1: Thank you for the opportunity to review the manuscript.

Title: Neural correlates of the improvement of cognitive performance resulting from enhanced sense of competence: a magnetoencephalography study

The present finding of study is that perception of an academic reward induces increase in motivation and task-related activity during processing of executive function. This paper is well written and addresses an important topic and provides a new concept of educational neuroscience. I think that the paper is acceptable for the publication of PLOS ONE. However, there are several concerns to be improved.

Finally, MEG data analysis might have be conducted in 15 participants. In this situation, the results of subjective motivation, task performance, and these correlations including the neural activations of right IFG (BA47) and left precuneus (BA7) should be mentioned and discussed. In Figure 5, it is difficult to identify or visualize the activated regions of this SPM map, and thus, I recommend use other versions of map such as a render map.

Reviewer #2: Comments to Editors/Decision:

Reject. Poor theoretical motivation. Methods details lacking. Analysis could be improved.

Comments to Authors:

Overall, the study seemed to not be motivated well from a theoretical point of view. It was not clear to the reader what the implications of these data were for the academic setting. The 1-back Stroop task is an interesting activation task. What is the justification for its use to evaluate motivation and academic performance? Has this task been validated previously using behavioral measures related to academic performance specifically?

Diversity & Inclusion concern: Why were only male subjects studied here? This was a study of healthy individuals. What is the justification for exclusion of females in this study? This is a major concern for a study such as this in the 21st century.

Omission of many critical methodological details. The manuscript was missing many critical details – the reader should not have to go hunting for them in another paper. [See my specific comments below.] Please also see the COBIDAS MEEG white paper for what details should be included in reporting methods for data acquisition and analysis for MEEG studies: Pernet P, Garrido M, Gramfort A, Maurits N, Michel C, Pang E, Salmelin R, Schoffelen JM, Valdes-Sosa PA, Puce A. (2018) Best practices in data analysis and sharing in neuroimaging using MEEG. https://osf.io/a8dhx/ And also see: Pernet P, Garrido M, Gramfort A, Maurits N, Michel C, Pang E, Salmelin R, Schoffelen JM, Valdes-Sosa PA, Puce A. (2020) Issues and recommendations from the OHBM COBIDAS MEEG committee for reproducible EEG and MEG research. Invited Perspective, Nature Neuroscience, 23(12):1473-1483. doi: https://10.1038/s41593-020-00709-0

Finally, I have a number of specific questions from reading the manuscript. Most of these are methods related:

1. Stimuli. ‘…one of four words (Red, Blue, Yellow, or Green in Japanese)…’ What was the stimuli? Were these Kanji pictograms or Kana characters etc? ‘…the color of the present word…’ Do you mean the color in which the word appeared in? What was the visual angle subtended by the stimuli (horizontal & vertical)?

2. Control task. ‘…image indicating the temperature of the fingers was the same for every participant…’ How was the image changed during the course of the task to make the control task seem real for the subject?

3. Stimulus presentation software details are missing. What software was used to present stimuli?

4. Data acquisition details are missing. What was the location of the reference coils? What was the low pass filter setting for data acquisition? High pass is reported, but low pass has been omitted.

5. How were eye movements & blinks monitored? These details have been omitted from the manuscript.

6. How were the fiducials digitized? Polhemus system? These details should be described in the manuscript.

7. Were ‘empty room’ recordings used to help reject non-brain artifacts? This should be explicitly stated.

8. ‘Epochs of the MEG data, including artifacts, were identified visually and excluded from the analysis.’ What artifacts exactly were excluded from the data? What was the total length of the data epochs? Was the pre-stimulus baseline 300 ms?

9. All version numbers of named software should be included in the analysis.

10. It is not exactly clear if individual structural MRIs were or were not performed in the subjects. Or was an MRI group template brain was used for the analysis? The description is not clear. If individual subject MRIs were performed in each subject & fiducials were digitized in the MRI images, was the MEG study performed prior to the MRI study to make sure that no additional magnetization was present?

11. Statistical analysis overall. It is not clear to me why a generalized overall analysis for main effects and their interactions was not performed in SPM. Main effects of say Task [Stroop, Control] and Iteration [1-4] should have first been performed before resorting to individual t-tests.

12. ‘The MEG data were superimposed on the MRI images using information obtained from

these markers and the MEG localization coils.’ So what method was used to achieve the co-registration?

13. The reporting of the data should include individual subject data points in the histograms & not just means. Effect sizes should be described together with all results in the text.

6. PLOS authors have the option to publish the peer review history of their article (what does this mean?). If published, this will include your full peer review and any attached files.

Reviewer #1: No

Reviewer #2: No

---

## [Author Response · Author response to Decision Letter 0]

12 Apr 2021

Responses to the comments

Reviewer 1

1. MEG data analysis might have be conducted in 15 participants. In this situation, the results of subjective motivation, task performance, and these correlations including the neural activations of right IFG (BA47) and left precuneus (BA7) should be mentioned and discussed. 

Response:

The MEG data from four participants were excluded because the numbers of the trials of their MEG data after the removal of the trials contaminated with artifacts were not sufficient for the analysis as described in the Results (Page 13, lines 7 to 11, in the revised manuscript) and therefore, the MEG data analysis have been conducted in 16 participants. Since we were interested in the neural activity related to the increased task performance caused by an academic reward such that induce a sense of competence, we performed the correlation analysis of the neural activity with the alteration of the task performance (i.e., the correction rate in this case) and discussed the supposed neural mechanisms by which task performance is enhanced by academic rewards based on the results of this correlation analysis. As the Reviewer 1 suggested, we added the correlation analysis of the neural activity with the alteration of the subjective motivation; however, we were not able to detect the alteration of the neural activity that correlated with the alteration of the subjective motivation caused by the academic reward. We have discussed the possibility that the neural activity related to the increase in motivation might have been missed in the present study in the Discussion (Page 16, lines 20 to 23, in the revised manuscript). We added the expression, “i.e., the MEG data were analyzed for 16 participants”, in the Results (Page 13, lines 7 to 8, in the revised manuscript). We added the sentences, “As an additional analysis, the correlation analysis of the neural activity with the alteration of the subjective motivation was performed: The statistical threshold for this additional analysis was also set at P < 0.025”, in the Materials and methods (Page 9, lines 2 to 4, in the revised manuscript) and “The alteration of the neural activity that correlated with the alteration of the subjective motivation were not observed”, in the Results (Page 13, lines 3 to 4, in the revised manuscript).

2. In Figure 5, it is difficult to identify or visualize the activated regions of this SPM map, and thus, I recommend use other versions of map such as a render map.

Response:

Thank you for the suggestion. We revised the Figure 5 as the Reviewer 1 suggested.

Reviewer 2

1. It was not clear to the reader what the implications of these data were for the academic setting.

Response:

Thank you for the useful comment. We added the sentence, “Our finding that cognitive performance can be increased by academic rewards through enhanced activation of task related brain regions would motivate further studies to clarify the neural mechanisms by which academic performance is improved by enhanced motivation to learn in longer time span, leading to establish scientific basis for the educational methods aimed to enhance motivation to learn and to devise effective educational methods”, in the Discussion (Page 17, lines 9 to 14, in the revised manuscript).

2. The 1-back Stroop task is an interesting activation task. What is the justification for its use to evaluate motivation and academic performance? Has this task been validated previously using behavioral measures related to academic performance specifically?

Response:

We needed a cognitive task with which the improvement of the task performance caused by academic rewards can be evaluated in our present study. We have been using a type of 2-back task as the cognitive task to induce mental fatigue in our past studies investigating the neural mechanisms of fatigue and have realized that it is difficult to evaluate the slight alteration of cognitive performance, such that caused by academic rewards, using by the task performance of the 2-back task. Therefore, we have developed and tried several cognitive tasks by ourselves and finally selected the 1-back Stroop task as the cognitive task which would be able to evaluate the slight alteration of cognitive performance caused by academic rewards. To the best of our knowledge, there have been no reports in which 1-back Stroop task was used to assess behavioral measures related to academic performance. Therefore, one of the aims of our present study was to examine whether the 1-back Stroop task is able to evaluate the alteration of cognitive performance caused by academic rewards. We added a sentence in the Introduction as follows: “The aim of this study was to examine whether the 1-back Stroop task we have developed is able to evaluate the alteration of cognitive performance caused by academic rewards and to clarify the neural correlates of the enhanced cognitive task performance resulting from academic rewards that induce a sense of competence” (Page 4, lines 7 to 10, in the revised manuscript).

3. Why were only male subjects studied here? This was a study of healthy individuals. What is the justification for exclusion of females in this study? This is a major concern for a study such as this in the 21st century.

Response:

The main reason that the participants of our present study were male was to minimize the magnetic noises caused by metallic components of cosmetics, accessories, closes, and so on. We have conducted several MEG studies with female participants because the purpose of these studies was to examine some kinds of neural activity in female participants and found that even though we have carefully announced to the participants that they should not wear makeup, accessories, and closes containing metallic components, including underwear, the MEG data from more than half of the female participants were contaminated with magnetic noises from unidentified sources, which were thought to be the metallic components included in cosmetics, accessories, closes, and so on: Unfortunately, since our MEG facility have no changing room and the space for removing and re-wearing makeup, we are not able to identify the source of the magnetic noises by removing potential sources of the magnetic noises. In contrast, it is rare for male participants to ware makeup, accessories, and closes containing metallic components, which are the potential sources of magnetic noise. Therefore, we decided to recruit male participants in this present study and planned to perform another study in the future including female participants to generalize our findings if our present experimental design was effective for clarifying the neural mechanisms by which cognitive performance is improved by academic rewards in male participants.

We understand the concerns raised by the Reviewer 2 and the editor that it is desirable to include female participants in the human studies in the 21st century. However, we do not have means to perform additional experiments with female participants at this time because the MEG experiments with human participants have been stopped in our university since April in 2020 due to the expansion of COVID-19 infection in Japan, and in addition, our funded project is finished.

We would like to point out that although our findings may not be generalized to the population as described in the Discussion (Page 16, lines 18 to 20, in the revised manuscript), we believe that our finding that cognitive performance can be increased by academic rewards through enhanced activation of task related brain regions is fundamental to establish scientific basis for the educational methods aimed to enhance motivation to learn and have great value to motivate further important studies such that clarify whether there are differences between male and female participants in the neural mechanisms of the improvement of cognitive performance caused by enhanced motivation to learn.

4. Stimuli. ‘…one of four words (Red, Blue, Yellow, or Green in Japanese)…’ What was the stimuli? Were these Kanji pictograms or Kana characters etc? ‘…the color of the present word…’ Do you mean the color in which the word appeared in? What was the visual angle subtended by the stimuli (horizontal & vertical)?

Response:

The words presented in Japanese were Kanji pictograms. We revised the description in the Materials and methods as follows: “Red, Blue, Yellow, or Green in Japanese Kanji pictograms” (Page 5, line 25 to page 6, line 1 and page 30, line 6, in the revised manuscript). We also revised the description, “the color of the present word”, to be “the font color of the word presented at the time” (Page 6, line 5 and page 30, lines 8 to 9, in the revised manuscript). The visual angle of the Kanji pictograms projected on the screen was 5.7° × 5.7° (horizontal × vertical). We added the description, “The visual angle of the Kanji pictograms projected on the screen was 5.7° × 5.7° (horizontal × vertical)”, in the Materials and methods (Page 6, lines 2 to 3, in the revised manuscript).

5. Control task. ‘…image indicating the temperature of the fingers was the same for every participant…’ How was the image changed during the course of the task to make the control task seem real for the subject?

Response:

The image indicating the temperature of the fingers was the same during the course of the task. In order to make the participants believe that the temperature of their fingers was actually measured, they were instructed that the bar indicating their temperature corresponded to some range of temperature although the scale and the unit were not shown and therefore, the image indicating the temperature of the fingers might be the same during the course of the experiment even if slight fluctuation of the temperature existed. We revised and added the descriptions in the Materials and methods: “However, in fact, the temperature of the fingers was not assessed during the experiments, and the image indicating the temperature of the fingers was the same for every participant and the same during the course of the task. In order to make the participants believe that the temperature of their fingers was actually measured, they were instructed that the bar indicating their temperature corresponded to some range of temperature although the scale and the unit were not shown and therefore, the image indicating the temperature of the fingers might be the same during the course of the experiment even if slight fluctuation of the temperature existed”, in the Materials and methods (Page 7, lines 6 to 13, in the revised manuscript).

6. Stimulus presentation software details are missing. What software was used to present stimuli?

Response:

Thank you for the suggestion. We added the sentence, “Visual stimuli were presented by using OpenSesame software (Version 3.1.7-py2.7-win32; Mathot et al., 2012)”, in the Materials and methods (Page 6, lines 12 to 13, in the revised manuscript).

Reference:

Mathot S, Schreij D, Theeuwes J. OpenSesame: an open-source, graphical experiment builder for the social sciences. Behavior research methods. 2012;44(2):314-24. Epub 2011/11/16. doi: 10.3758/s13428-011-0168-7. PubMed PMID: 22083660; PubMed Central PMCID: PMC3356517.

7. Data acquisition details are missing. What was the location of the reference coils? What was the low pass filter setting for data acquisition? High pass is reported, but low pass has been omitted.

Response:

Thank you for the suggestions. The reference sensors were located in the liquid helium dewar, away from participants’ head, in the magnetically shielded room. The original MEG data were high-pass filtered at 0.3 Hz and low-pass filtered at 500 Hz. We added the sentence, “The reference sensors were located in the liquid helium dewar, away from participants’ head, in the magnetically shielded room” in the Materials and methods (Page 8, lines 8 to 10, in the revised manuscript). We revised the sentence regarding the filter setting for data acquisition to be “The sampling rate was 1,000 Hz and the obtained data were high-pass filtered at 0.3 Hz and low-pass filtered at 500 Hz” (Page 8, lines 1 to 2, in the revised manuscript).

8. How were eye movements & blinks monitored? These details have been omitted from the manuscript.

Response:

Thank you for the suggestion. We added the description, “In addition, electro-oculography (EOG) was recorded, from electrodes placed over the left and right ends of the eyebrow of the left eye by using Neurofax 1000 (Nihon-Kohden, Tokyo, Japan) with sampling rate of 1000 Hz applying 0.5 Hz low-cut filter and 120 Hz high-cut filter, over the MEG recordings to help identify the physiological noises caused by eye blinks and eye movements”, in the Materials and methods (Page 8, lines 14 to 19, in the revised manuscript).

9. How were the fiducials digitized? Polhemus system? These details should be described in the manuscript.

Response:

We did not use specific fiducials-digitize system. As described in the Materials and methods (Page 10, lines 11 to 18, in the revised manuscript), five marker coils were attached to the scalp before the start of the MEG recording and the magnetic field generated by each marker coil was measured during the MEG recording: The position of each marker coil was determined off-line after the recording using a software provided by the supplier of the MEG system (MEG 160; Yokogawa Electric Corporation). During the recording of MR image, five markers, which are detectable in the MR image, were attached to the scalp at just the same points as those at which marker coils were attached during the MEG recording. Then, the MEG data were co-registered onto the MR image based on the locations of these MEG marker coils and MRI markers by using the MEG 160 software. We revised the corresponding description in the Materials and methods to be “The MEG data were superimposed on the MR images using information obtained from these markers and the MEG localization coils, which were attached to the scalp over the MEG recording at just the same points as those at which MR-markers were attached, using the MEG 160 software” (Page 10, lines 15 to 18, in the revised manuscript). 

10. Were ‘empty room’ recordings used to help reject non-brain artifacts? This should be explicitly stated.

Response:

The data obtained by empty room recordings were not used in the present study. We added the description, “The data obtained by empty room recordings were not used”, in the Materials and methods (Page 8, lines 19 to 20, in the revised manuscript).

11. ‘Epochs of the MEG data, including artifacts, were identified visually and excluded from the analysis.’ What artifacts exactly were excluded from the data? What was the total length of the data epochs? Was the pre-stimulus baseline 300 ms?

Response:

Thank you for the suggestion. We revised the descriptions in the Materials and methods: “Epochs of the MEG data, including artifacts, such that originated from outside the magnetically shielded room remaining after the elimination using reference sensors and physiological noises caused by eye blinks and so on, were identified visually and excluded from the analysis. The total length of the epochs was 250 ms from -300 to -50 ms before the button press as described below” (Page 8, lines 10 to 14, in the revised manuscript).

12. All version numbers of named software should be included in the analysis.

Response:

Thank you for the suggestion. We added version numbers to all the software described in the manuscript (Page 6, lines 12 to 13; Page 9, line 6; Page 9, line 7).

13. It is not exactly clear if individual structural MRIs were or were not performed in the subjects. Or was an MRI group template brain was used for the analysis? The description is not clear. If individual subject MRIs were performed in each subject & fiducials were digitized in the MRI images, was the MEG study performed prior to the MRI study to make sure that no additional magnetization was present?

Response:

Individual structural MRIs were performed for each participant. We revised the description regarding MR imaging in the Materials and methods to be “anatomical magnetic resonance imaging (MRI) was performed for each participant using a Philips Achieva 3.0 TX scanner to permit registration of magnetic source locations with their respective anatomical locations” (Page 10, lines 8 to 11, in the revised manuscript). Based on the empirical evidence reported in the literature (Gross et al., 2013) suggesting that the negative effects caused by an MRI scan on MEG data typically disappear within 3 days after the MRI scan, the MEG study was planned to be performed at least 4 days after the MRI study to avoid effects caused by magnetization if the MRI study were not able to be performed before the MEG study. In fact, the interval between the MEG study and the MRI scan was more than 5 days in our present study when the MRI scan preceded the MEG study. We added the description, “Based on the empirical evidence reported in the literature (Gross et al., 2013) suggesting that the negative effects caused by an MRI scan on MEG data typically disappear within 3 days after the MRI scan, the MEG study was planned to be performed at least 4 days after the MRI study to avoid effects caused by magnetization if the MRI study were not able to be performed before the MEG study. In fact, the interval between the MEG study and the MRI scan was more than 5 days in our present study when the MRI scan preceded the MEG study”, in the Materials and methods (Page 10, lines 18 to 24, in the revised manuscript).

Reference:

Gross J, Baillet S, Barnes GR, Henson RN, Hillebrand A, Jensen O, et al. Good practice for conducting and reporting MEG research. Neuroimage. 2013;65:349-63. doi: 10.1016/j.neuroimage.2012.10.001. PubMed PMID: WOS:000312283900032.

14. Statistical analysis overall. It is not clear to me why a generalized overall analysis for main effects and their interactions was not performed in SPM. Main effects of say Task [Stroop, Control] and Iteration [1-4] should have first been performed before resorting to individual t-tests.

Response:

Since we were interested in the neural activity correlated to the increase of the cognitive performance caused by academic rewards, we performed correlation analysis of the MEG data with the improvement of the task performance. Therefore, we performed correlation analyses rather than individual t-tests in the SPM analyses in our present study.

15. ‘The MEG data were superimposed on the MRI images using information obtained from these markers and the MEG localization coils.’ So what method was used to achieve the co-registration?

Response:

Please refer to the Response to the Reviewer 2’s comment #9.

16. The reporting of the data should include individual subject data points in the histograms & not just means. Effect sizes should be described together with all results in the text.

Response:

As the Reviewer 2 suggested, we revised Figure 2 and Figure 3 to show individual data points and the effect sizes were added in the Results (Page 12, line 4; Page 12, line 7; Page 12, line 10; Page 12, line 17; Page 12, line 20; Page 12, line 22). We added the description, “Effect size of each paired t-test was calculated by using G*Power software (Version 3. 1. 9. 2; Faul et al., 2007)”, in the Materials and methods (Page 11, lines 8 to 9, in the revised manuscript).

Reference:

Faul F, Erdfelder E, Lang AG, Buchner A. G*Power 3: a flexible statistical power analysis program for the social, behavioral, and biomedical sciences. Behavior research methods. 2007;39(2):175-91. Epub 2007/08/19. PubMed PMID: 17695343.

END

---

## [Decision Letter · Decision Letter 1]

18 May 2021

PONE-D-21-01757R1

Neural correlates of the improvement of cognitive performance resulting from enhanced sense of competence: a magnetoencephalography study

PLOS ONE

Dear Dr. Ishii,

Thank you for submitting your manuscript to PLOS ONE. After careful consideration, we feel that it has merit but does not fully meet PLOS ONE’s publication criteria as it currently stands. Therefore, we invite you to submit a revised version of the manuscript that addresses the points raised during the review process.

We look forward to receiving your revised manuscript.

Kind regards,

Desmond J. Oathes

Academic Editor

PLOS ONE

Journal Requirements:

Additional Editor Comments (if provided):

Although the new reviewer recommended rejecting the manuscript, the critiques seem to me to be potentially addressable.

Reviewers' comments:

Reviewer's Responses to Questions

**Comments to the Author**

1. If the authors have adequately addressed your comments raised in a previous round of review and you feel that this manuscript is now acceptable for publication, you may indicate that here to bypass the “Comments to the Author” section, enter your conflict of interest statement in the “Confidential to Editor” section, and submit your "Accept" recommendation.

Reviewer #1: (No Response)

Reviewer #3: (No Response)

2. Is the manuscript technically sound, and do the data support the conclusions?

Reviewer #1: Yes

Reviewer #3: Partly

3. Has the statistical analysis been performed appropriately and rigorously? 

Reviewer #1: Yes

Reviewer #3: I Don't Know

4. Have the authors made all data underlying the findings in their manuscript fully available?

Reviewer #1: Yes

Reviewer #3: Yes

5. Is the manuscript presented in an intelligible fashion and written in standard English?

Reviewer #1: Yes

Reviewer #3: Yes

6. Review Comments to the Author

Reviewer #1: (No Response)

Reviewer #3: The results of this study were not strong enough and the number of participants was not big enough, therefore the significance of this work is limited. Personally, I think the results of cortical source level study based on MEG data is doubtful. I prefer sensor level analysis of MEG. The brain activity could be studied by analyzing the sensor level event related response.

1. In page 10, the logical of this sentence, “……caused by magnetization if the MRI study were not able to be performed before the MEG study,” is wrong.

2. In the result section, I don’t know why authors used the description of “paired t-test without/with the Bonferroni correction”. The Bonferroni correction itself should be performed equally to all the significant test of one study.

3. Since, two conditions were performed by all participances. It is difficult to say this kind of experiment design will induce bias. The participance may intend to probe the feedback of “finger temperature and competence of their work” due to the curiosity of why two different experiment conditions they faced with. Which condition was first phase of the experiment of each participants may also affect the outcome of their performance.

4. It is also difficult to ensure the improvement of cognitive performance was not the result of the attention which arise by the performance relate feedback. Current control condition is not performance related condition, any performance related feedback may induce similar results like target condition.

5. Authors should also perform sensor level analysis of MEG data to confirm the relation between enhance sense of competence and the improvement of cognitive performance by neural rhythms.

7. PLOS authors have the option to publish the peer review history of their article (what does this mean?). If published, this will include your full peer review and any attached files.

Reviewer #1: No

Reviewer #3: **Yes: **DUAN Fang

---

## [Author Response · Author response to Decision Letter 1]

4 Jun 2021

Responses to the comments

Reviewer 3

1. In page 10, the logical of this sentence, “……caused by magnetization if the MRI study were not able to be performed before the MEG study,” is wrong.

Response:

Thank you for the suggestion. We revised the description to be “if the MEG study were not able to be performed before the MRI study” (Page 10, line 23, in the revised manuscript). 

2. In the result section, I don’t know why authors used the description of “paired t-test without/with the Bonferroni correction”. The Bonferroni correction itself should be performed equally to all the significant test of one study.

Response:

We primarily tested whether the alterations of the level of cognitive performance between task 4 and task 1 were observed in the target or control conditions. In this case, since the paired t-test was performed two times (i.e., for the target condition and for the control condition), the Bonferroni correction was performed. For the additional analysis to confirm the primary analysis, we compared the level of the alteration of correction rate caused between task 4 and task 1 in the target condition with that in the control condition. In this additional analysis, since the paired t-test was performed once, therefore the Bonferroni correction was not performed. Likewise, the analysis which examined whether the alterations of the level of motivation between task 4 and task 1 were observed in the target or control conditions was performed with Bonferroni correction and the analysis in which the alteration of the level of motivation caused between task 4 and task 1 in the target condition was compared with that in the control condition was performed without the Bonferroni correction.

3. Since, two conditions were performed by all participances. It is difficult to say this kind of experiment design will induce bias. The participance may intend to probe the feedback of “finger temperature and competence of their work” due to the curiosity of why two different experiment conditions they faced with. Which condition was first phase of the experiment of each participants may also affect the outcome of their performance.

Response:

In our present study, the half of the participants performed the target condition first and the other half of the participants performed the control condition first (i.e., the crossover design). While the crossover design has several merits such that being able to avoid problems of comparability of target and control conditions with regard to between-person confounding variables, as the Reviewer 3 suggested, the effects of the academic reward on the cognitive performance might have not been completely the same between the participants performed the target condition first and those performed control condition first; however, since the half of the participants performed the target condition first (group 1) and the other half of the participants performed the control condition first (group 2), our finding that the decrease of the correction rate in the control condition was greater than that in the target condition was not because the target condition was performed first or because the control condition was performed first. In fact, the decrease of the correction rate in the control condition was greater than that in the target condition in both of the groups (the decrease of the correction rate in the control condition over that in the target condition in the group 1 and group 2; mean ± SD: 4.73 ± 5.73 and 1.04 ± 5.80, respectively).

4. It is also difficult to ensure the improvement of cognitive performance was not the result of the attention which arise by the performance relate feedback. Current control condition is not performance related condition, any performance related feedback may induce similar results like target condition.

Response:

We intended to test whether the improvement of cognitive performance would be caused by the performance related feedback which would work as an academic reward. Therefore, we did not examine whether the improvement of cognitive performance would be caused by the performance related feedback which would not work as academic rewards such that showing images indicating that the performance of each individual in the 1-back Stroop task was equal to or below the average. Since it was shown that the improvement of cognitive performance can be caused by the performance related feedback which would work as an academic reward in our present study, it is of interest to examine whether the improvement of cognitive performance is also caused by showing images indicating that the performance of each individual is equal to or below the average and to clarify the difference in the neural mechanisms that enhance cognitive performance between the condition with performance related feedback which would work as an academic reward and that would not if the cognitive performance is also enhanced by the performance related feedback which would not work as an academic reward in future researches. We added the descriptions, “Since we intended to test whether the improvement of cognitive performance would be caused by the performance related feedback which would work as an academic reward that induce a sense of competence, we did not examine whether the improvement of cognitive performance would be caused by the performance related feedback which would not work as academic rewards such that showing images indicating that the performance of each individual in the 1-back Stroop task was equal to or below the average in our present study. It is of interest to examine whether the improvement of cognitive performance is also caused by showing images indicating that the performance of each individual is equal to or below the average in future researches”, in the Discussion (Page 17, lines 5 to 14, in the revised manuscript).

5. (Personally, I think the results of cortical source level study based on MEG data is doubtful. I prefer sensor level analysis of MEG.) Authors should also perform sensor level analysis of MEG data to confirm the relation between enhance sense of competence and the improvement of cognitive performance by neural rhythms.

Response:

Since we wanted to clarify precise brain regions correlated with the enhanced cognitive task performance resulting from academic rewards, we planned to perform source reconstruction of the MEG data in our present study. We understand the usefulness of the sensor level analysis of MEG data and that the source reconstruction of the MEG data involves solving ill-posed inverse problem; however, the source level analysis of MEG data is sensitive to the head position in the whole-head MEG helmet relative to the MEG sensors. In our present study, the position of the head in the MEG helmet was not precisely the same between the participants and even within each participant between the conditions because the participants’ heads were outside of the MEG helmet during the rest period between the conditions. Therefore, we think that the MEG analyses with source reconstruction is adequate in our present study. In fact, there are studies reporting that the reliability of the MEG analyses with source reconstruction were comparable or superior to that of the sensor level analyses (for example: Lu et al., 2007; Tan et al., 2015, 2016; Rodriguez-Gonzalez, 2020). We added the description, “We performed source reconstruction of the MEG data because we wanted to clarify precise brain regions correlated with the enhanced cognitive task performance resulting from academic rewards. Since the sensor level analysis of MEG data is sensitive to the head position in the whole-head MEG helmet and the position of the head in the MEG helmet was not precisely the same between the participants and even within each participant between the conditions because the participants’ heads were out of the MEG helmet during the rest period between the conditions in our present study, we think that the MEG analysis with source reconstruction is adequate in our present study”, in the Discussion (Page 17, lines 14 to 22, in the revised manuscript). 

We also revised the sentence, “A 5-min rest period was instituted between the target and control conditions”, to be, “A 5-min rest period, during which participants’ heads were outside of the MEG helmet, was instituted between the target and control conditions”, in the Materials and methods (Page 7, lines 15 to 17, in the revised manuscript).

References:

1. Lu BY, Edgar JC, Jones AP, Smith AK, Huang MX, Miller GA, et al. Improved test-retest reliability of 50-ms paired-click auditory gating using magnetoencephalography source modeling. Psychophysiology. 2007;44(1):86-90. Epub 2007/01/24. doi: 10.1111/j.1469-8986.2006.00478.x. PubMed PMID: 17241143.

2. Tan HR, Gross J, Uhlhaas PJ. MEG-measured auditory steady-state oscillations show high test-retest reliability: A sensor and source-space analysis. Neuroimage. 2015;122:417-26. Epub 2015/07/29. doi: 10.1016/j.neuroimage.2015.07.055. PubMed PMID: 26216274.

3. Tan HM, Gross J, Uhlhaas PJ. MEG sensor and source measures of visually induced gamma-band oscillations are highly reliable. Neuroimage. 2016;137:34-44. Epub 2016/05/08. doi: 10.1016/j.neuroimage.2016.05.006. PubMed PMID: 27153980; PubMed Central PMCID: PMCPMC5405052.

4. Rodriguez-Gonzalez V, Gomez C, Shigihara Y, Hoshi H, Revilla-Vallejo M, Hornero R, et al. Consistency of local activation parameters at sensor- and source-level in neural signals. J Neural Eng. 2020;17(5). doi: ARTN 056020

10.1088/1741-2552/abb582. PubMed PMID: WOS:000579634100001.

END

---

## [Decision Letter · Decision Letter 2]

29 Jun 2021

PONE-D-21-01757R2

Neural correlates of the improvement of cognitive performance resulting from enhanced sense of competence: a magnetoencephalography study

PLOS ONE

Dear Dr. Ishii,

Thank you for submitting your manuscript to PLOS ONE. After careful consideration, we feel that it has merit but does not fully meet PLOS ONE’s publication criteria as it currently stands. Therefore, we invite you to submit a revised version of the manuscript that addresses the points raised during the review process.

To be considered for publication, please provide the details and rationale requested by Reviewer #3 for the source analyses.

We look forward to receiving your revised manuscript.

Kind regards,

Desmond J. Oathes

Academic Editor

PLOS ONE

Journal Requirements:

Reviewers' comments:

Reviewer's Responses to Questions

**Comments to the Author**

1. If the authors have adequately addressed your comments raised in a previous round of review and you feel that this manuscript is now acceptable for publication, you may indicate that here to bypass the “Comments to the Author” section, enter your conflict of interest statement in the “Confidential to Editor” section, and submit your "Accept" recommendation.

Reviewer #1: All comments have been addressed

Reviewer #3: All comments have been addressed

2. Is the manuscript technically sound, and do the data support the conclusions?

Reviewer #1: Yes

Reviewer #3: Partly

3. Has the statistical analysis been performed appropriately and rigorously? 

Reviewer #1: Yes

Reviewer #3: I Don't Know

4. Have the authors made all data underlying the findings in their manuscript fully available?

Reviewer #1: Yes

Reviewer #3: Yes

5. Is the manuscript presented in an intelligible fashion and written in standard English?

Reviewer #1: Yes

Reviewer #3: Yes

6. Review Comments to the Author

Reviewer #1: (No Response)

Reviewer #3: Author answer most of my questions. But on the issue of source level study, as I said, in my opinion the results of cortical source level study based on MEG data is doubtful. Clarify the precise brain region is a good will, however 160 channels of MEG may be can not provide adequate information to reconstruct source level estimation. If authors insist on source level, at least, they should select proper parameters to ensure they could avoid ill-pose issue during the source level studies. Therefore, I suggest authors descript detail of the parameter setting of source level study. And discuss on the aspect of why they set those parameters, only based on other references? Or based on the theory of source reconstruction.

7. PLOS authors have the option to publish the peer review history of their article (what does this mean?). If published, this will include your full peer review and any attached files.

Reviewer #1: No

Reviewer #3: No

---

## [Author Response · Author response to Decision Letter 2]

7 Jul 2021

Response to the comment

Reviewer 3

1. Author answer most of my questions. But on the issue of source level study, as I said, in my opinion the results of cortical source level study based on MEG data is doubtful. Clarify the precise brain region is a good will, however 160 channels of MEG may be can not provide adequate information to reconstruct source level estimation. If authors insist on source level, at least, they should select proper parameters to ensure they could avoid ill-pose issue during the source level studies. Therefore, I suggest authors descript detail of the parameter setting of source level study. And discuss on the aspect of why they set those parameters, only based on other references? Or based on the theory of source reconstruction.

Response:

We used an adaptive spatial filtering method implemented in the software (Brain Rhythmic Analysis for MEG software, BRAM) provided by the supplier of the MEG system (Yokogawa Electric Corporation, Tokyo, Japan) to perform source reconstruction of the MEG data, as described in our previous manuscript. The adaptive spatial filtering method that implemented in the software was a lead-field normalized version of a linearly constrained minimum variance filtering method (Van Veen et al., 1997) and the procedure was optimized for time-frequency source reconstruction (Dalal et al., 2008). The mathematical details of the source reconstruction performed in our present study are present in the previous report (Ueno et al., 2012). The parameter settings of the software were as follows: The active period was “-300 to -50 ms” before the button press in task 4, the baseline period was “-300 to 500 ms” before the button press in task 1, the functional type was {“ERD”, “ERS”}, the functional unit was “F-ratio [dB]”, the voxel spacing was “5 mm”, and the voxel range was “inside the spherical conductor model”.

Although, as the Reviewer 3 suggested, the resolution and the accuracy of source reconstructions increases as the number of MEG channels increases (Vrba et al, 2004), it has been theoretically shown that the minimum-variance spatial filter with normalized lead-field has no location bias and this finding was confirmed by the numerical experiments which simulated an MEG system with 148 channels (Sekihara et al., 2005). It has also been demonstrated that the lead-field normalized minimum-variance spatial filter applied to the MEG data recorded by using Yokogawa MEG system (i.e., 160 channels) had high spatial resolution (Sekihara et al., 2005).

We revised the descriptions in the Materials and methods to include the details of the source reconstruction: “The adaptive spatial filtering method that implemented in the software was a lead-field normalized version of a linearly constrained minimum variance filtering method (Van Veen et al., 1997) and the procedure was optimized for time-frequency source reconstruction (Dalal et al., 2008). The mathematical details of the source reconstruction performed in our present study are present in the previous report (Ueno et al., 2012). The voxel size was set at 5.0 � 5.0 � 5.0 mm. Under both the target and control conditions, the oscillatory power of the MEG data for task 4 in a 250-ms time window (from –300 to –50 ms before the button press) was calculated relative to that of task 1. The parameter settings of the software were as follows: The active period was “-300 to -50 ms” before the button press in task 4, the baseline period was “-300 to 500 ms” before the button press in task 1, the functional type was {“ERD”, “ERS”}, the functional unit was “F-ratio [dB]”, the voxel spacing was “5 mm”, and the voxel range was “inside the spherical conductor model”” (Page 9, lines 3 to 15, in the revised manuscript).

References:

1. Van Veen BD, van Drongelen W, Yuchtman M, Suzuki A. Localization of brain electrical activity via linearly constrained minimum variance spatial filtering. IEEE Trans Biomed Eng. 1997;44(9):867-80. Epub 1997/09/01. doi: 10.1109/10.623056. PubMed PMID: 9282479.

2. Dalal SS, Guggisberg AG, Edwards E, Sekihara K, Findlay AM, Canolty RT, et al. Five-dimensional neuroimaging: localization of the time-frequency dynamics of cortical activity. Neuroimage. 2008;40(4):1686-700. Epub 2008/03/22. doi: 10.1016/j.neuroimage.2008.01.023. PubMed PMID: 18356081; PubMed Central PMCID: PMC2426929.

3. Ueno S, Okumura E, Remijn GB, Yoshimura Y, Kikuchi M, Shitamichi K, et al. Spatiotemporal frequency characteristics of cerebral oscillations during the perception of fundamental frequency contour changes in one-syllable intonation. Neurosci Lett. 2012;515(2):141-6. Epub 2012/04/03. doi: 10.1016/j.neulet.2012.03.031. PubMed PMID: 22465137.

4. Vrba J, Robinson SE, McCubbin J. How many channels are needed for MEG? Neurol Clin Neurophysiol. 2004;2004:99. Epub 2005/07/14. PubMed PMID: 16012656.

5. Sekihara K, Sahani M, Nagarajan SS. Localization bias and spatial resolution of adaptive and non-adaptive spatial filters for MEG source reconstruction. Neuroimage. 2005;25(4):1056-67. Epub 2005/04/27. doi: 10.1016/j.neuroimage.2004.11.051. PubMed PMID: 15850724; PubMed Central PMCID: PMC4060617.

END

---

## [Editor Report · Decision Letter 3]

14 Jul 2021

Neural correlates of the improvement of cognitive performance resulting from enhanced sense of competence: a magnetoencephalography study

PONE-D-21-01757R3

Dear Dr. Ishii,

We’re pleased to inform you that your manuscript has been judged scientifically suitable for publication and will be formally accepted for publication once it meets all outstanding technical requirements.

Kind regards,

Desmond J. Oathes

Academic Editor

PLOS ONE
---

## [Editor Report · Acceptance letter]

16 Jul 2021

PONE-D-21-01757R3 

Neural correlates of the improvement of cognitive performance resulting from enhanced sense of competence: a magnetoencephalography study 

Dear Dr. Ishii:

I'm pleased to inform you that your manuscript has been deemed suitable for publication in PLOS ONE. Congratulations! Your manuscript is now with our production department. 

Kind regards, 

on behalf of

Dr. Desmond J. Oathes 

Academic Editor

PLOS ONE